# Transitional Uncertainty with Intermediate Neural Gaussian Processes

## Abstract

In this paper, we discuss feature engineering for single-pass uncertainty estimation. For accurate uncertainty estimates, neural networks must extract differences in the feature space that quantify uncertainty. This could be achieved by current single-pass approaches that maintain feature distances between data points as they traverse the network. While initial results are promising, maintaining feature distances within the network representations frequently inhibits information compression and opposes the learning objective. We study this effect theoretically and empirically to arrive at a simple conclusion: preserving feature distances in the output is beneficial when the preserved features contribute to learning the label distribution and act in opposition otherwise. We then propose *Transitional Uncertainty with Intermediate Neural Gaussian Processes* (`TUrING Processes`) as a simple approach to address the shortcomings of current single-pass estimators. Specifically, we implement feature preservation by extracting features from intermediate representations before information is collapsed by subsequent layers. We refer to the underlying preservation mechanism as *transitional feature preservation*. We show that `TUrING Processes` match or outperform current single-pass methods on standard benchmarks and in practical settings where these methods are less reliable (imbalances, complex architectures, medical modalities).

## 1 Introduction

Effective single-pass uncertainty estimation in deep learning is governed by two design principals. The first is defining an output score that reflects uncertainty. For instance, we can measure uncertainty through distance from the training data (Liu et al., 2020), or the softmax confidence of the output (Mukhoti et al., 2023). While score design plays a crucial role in measuring uncertainty, the choice is dictated by application and uncertainty characteristics (Kendall & Gal, 2017). The second principle concerns information availability, namely whether the network can preserve features that reflect uncertainty information and does not "collapse" uncertain data points to certain representations (Van Amersfoort et al., 2020). We refer to the latter principle as *feature preservation*.

Despite their critical importance, preserving features is not trivial in neural networks. In particular, information compression is a desirable property of neural networks and a central component of the learning problem (Tishby et al., 2000). In spite of this discrepancy, current single-pass uncertainty methods preserve features by maintaining distances between data points in the output and risk inhibiting compression of application irrelevant information (Liu et al., 2020; Van Amersfoort et al., 2020; van Amersfoort et al., 2021; Mukhoti et al., 2023; Kwon et al., 2020; Prabhushankar & AlRegib). In common practical scenarios such as distributional shift, this characteristic manifests in performance decline (Postels et al., 2022). We provide a simple illustration of this effect in Figure 1a. The left plot shows the 2D neural network output features of two clusters when trained without explicit feature preservation. The network collapses the class clusters to single points creating a challenging setting for uncertainty estimation. The center plot shows the same 2D classification problem, but depicts a network trained with feature preservation constraints on the output. While the representations reflect uncertainty more accurately, the constraints introduce noise and cause an overlap of both clusters (signified by misclassified samples). In Section 3, we delve deeper into this effect and identify it as a limiting issue for current single-pass estimators both theoretically and empirically. These disadvantages motivate the search for alternative feature preservation approaches.

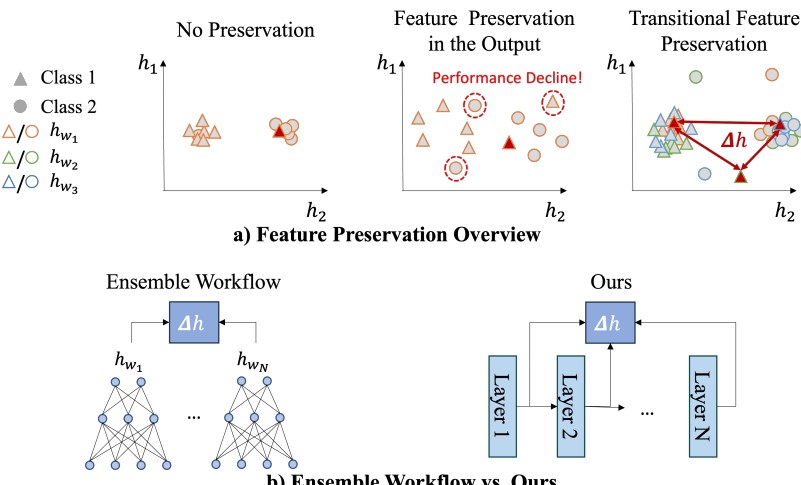

Figure 1: **a)** overview of different feature preservation paradigms. We show 2D representations of neural networks where $h_1$ and $h_2$ denote the output dimensions in the feature space. Left: conventional neural network. Samples are collapsed to two tight clusters with little uncertainty information. Center: feature preservation in the output. Feature differences are maintained resulting in higher uncertainty related content but also in a cluster overlap. Right: transitional feature preservation. uncertainty is measured from differences between several representations of the same sample (denoted as $h_{w_{1,2,3}}$). **b)** overview of our method workflow in comparison to ensembles. For ensembles, the representations come from several neural networks. Our method combines intermediate layer representations.

To this end, a more conservative strategy preserves features in the collection of multiple representations of the same sample (Malinin & Gales, 2018). From a preservation perspective, the approach is preferred as feature distances are encoded in differences (e.g. spread) of the individual sample representations and does not require constraining the compression property. We term this approach as *transitional feature preservation* and illustrate a toy example in the right plot of Figure 1a. We show three different representations of the same two clusters collected from different sources $h_{w_{1,2,3}}$. While the sources collapse points individually, the inter-source difference for a given sample reflects uncertainty (signified by $\Delta h$). A classic model that enjoys this property are deep ensembles (Lakshminarayanan et al., 2017), where each source representation is collected from a separate network. However, the evaluation requires several forward passes (one per network) and is limiting in applications with time and space constraints.

In this paper, we address shortcomings of current single-pass uncertainty estimators by implementing transitional feature preservation in a single forward pass. We contrast our method to ensembles in Figure 1b. Instead of sourcing multiple representations from separate networks, we utilize intermediate representations to extract features before they are collapsed by subsequent layers. We further combine our preservation approach with a single-pass uncertainty estimator, approximate Gaussian Processes, and arrive at a new single-pass model: *Transitional Uncertainty with Intermediate Neural Gaussian Processes* or `TUrING Processes` in short. Our estimator requires less labeled training data and outperforms current single-pass estimators both on standard benchmarks and other data modalities (CT scans). Further, we show that `TUrING Processes` are preferable under distributional shifts, complex architectures, and class imbalance; challenging settings for current single-pass estimators.

## 2 BACKGROUND

### 2.1 NEURAL NETWORKS IN THE INFORMATION PLANE

Consider the input space $\mathcal{X}$ with a corresponding probabilistic random variable $X$. Further, let $\mathcal{Y}$ denote a lower dimensional target space characterized by variable $Y$. The learning problem for

neural networks is equivalent to finding the minimally sufficient statistical mapping $h^*(X)$ with respect to the mutual information $I(X;Y)$ Shwartz-Ziv & Tishby (2017).

$$h^*(X) = \underset{h_w:I(h_w(X);Y)=I(X;Y)}{\arg\min} I(h_w(X);X) \tag{1}$$

Equation 1 is intuitive. During training, we optimize the network $h_w$ to fit the lower dimensional distribution $Y$ - i.e. maximize the mutual information $I(h_w(X);Y)$ between the representation distribution $h_w(X)$ and the target distribution $Y$. At the same time, the neural network must compress information irrelevant to the lower dimensional target variable $Y$. In Equation 1, we minimize the mutual information $I(h_w(X);X)$ between the representation $h_w(X)$ and the input distribution $X$. In practice, we derive $h_w$ from training data $D = \{y_i, x_i\}_{i=1}^N$ often collected from a subset of the full input space $\mathcal{X}_{ID} \subset \mathcal{X}$. As a result, the network optimizes with respect to the in-distribution variable $X_{ID}$ and produces arbitrarily bad results when exposed to out-of-distribution (OOD) data $\mathcal{X}_{OOD} \subset \mathcal{X} : \mathcal{X}_{OOD} \cap \mathcal{X}_{ID} = \emptyset$. For this reason, accurate uncertainty estimation is contingent on *modeling information related to the full input distribution* without over-fitting to $\mathcal{X}_{ID}$ (Liu et al., 2020).

## 2.2 DISTANCE-BASED FEATURE PRESERVATION IN THE OUTPUT

An intuitive approach to uncertainty estimation involves modeling distributional information in the output of the neural network feature extractor (Van Amersfoort et al., 2020; van Amersfoort et al., 2021; Mukhoti et al., 2023; Liu et al., 2020). The approach is practical as we can compute the network output in a single forward pass and measure uncertainty from the logits directly. To ensure accurate uncertainty estimates, current single-pass methods model input information by maintaining the distances between data points as they traverse the network. By preserving meaningful distances, we can estimate uncertainty by measuring the distance to the training domain $\mathcal{X}_{ID}$ (Liu et al., 2020). More formally, given an input space $\mathcal{X}$ equipped with a meaningful distance $d_X$, we learn a neural network $h_w : \mathcal{X} \rightarrow \mathcal{H}$ that allows a distance $d_H$ within the feature manifold that reflects the true distance $d_X$ (Liu et al., 2020):

$$d_H(h_w(\mathbf{x}_1), h_w(\mathbf{x}_2)) = d_X(\mathbf{x}_1, \mathbf{x}_2) \tag{2}$$

Unfortunately, neural networks do not naturally implement distance preservation and "collapse" data points to the same output. For this purpose, current single-pass methods enforce distance preservation artificially through constraints on the network representation. Popular examples of constraints include the two-sided gradient penalty (Gulrajani et al., 2017) and spectral normalization in combination with residual connections (Miyato et al., 2018).

## 3 MOTIVATION FOR TRANSITIONAL FEATURE PRESERVATION

### 3.1 THEORETICAL PITFALLS OF FEATURE PRESERVATION IN THE OUTPUT

While distance preservation in the output is a desirable property for uncertainty estimation, enforcing Equation 2 frequently results in performance degradation in neural networks: directly preserving distances in the network output can inhibit compression of information; a learning objective according to Equation 1. In this section, we provide theoretical justification that enforcing distance preservation on the network can act in direct opposition to the learning problem. We further arrive at an intuitive conclusion: distance preservation in the output is beneficial only when the preserved distances contain information related to the label distribution $Y$ - i.e. when they are relevant to the application.

We start our discussion by connecting the learning problem in Equation 1 to distances in the feature space. In particular, the minimally sufficient statistic shares the following dependency to feature distances for networks preserving distances in the output:

$$h^*(X) = \underset{h_w:\{I(f_H^k(h_w(M^k));Y^k)=I(M^k;Y^k),\ k\in[1,N_p]\}}{\arg\min} \sum_k I(f_H^k(h_w(M^k)); f_X^k(M^k)). \tag{3}$$

Here, $M^k$ is the corresponding random variable of a subset of the input space $\mathcal{M}^k \subset \mathcal{X}$, where each point in $\mathcal{M}^k$ has a unique distance to a fixed anchor point $\mathbf{x}_k \in \mathcal{X}$. Together, all $N_p$ partition subsets

form the entire input space $\mathcal{X} = \bigcup_{k \in [1, N_p]} \mathcal{M}^k : \bigcap_{k \in [1, N_p]} \mathcal{M}^k = \emptyset$. Further, $f_L^k(.) = d_L(\mathbf{x}_k; .)$ is a distance function with respect to the anchor point. We provide a formal definition of unique distance sets $\mathcal{M}^k$, as well as proof for Equation 3 in Appendix A.1.

Equation 3 is conceptually important as it provides a direct dependency between the learning problem of neural networks and distances in the feature plane. In particular, we note that the compression objective involves minimizing $I(d_H(h_w(\mathbf{x}_k); h_w(M^k)); d_X(\mathbf{x}_k; M^k))$ which is in direct opposition to preservation constraints that aim to maximize the similarities between the input and feature distances. We further note the maximization objective between the feature distances and the label distribution $I(f_H^k(h_w(M^k)); Y^k)$. Here, preservation constraints can be beneficial: when additional distances are preserved that contain information related to the label distribution $Y^k$, the term is increased. We arrive at the following observation:

**Observation 1.** *Preserving distances in the output is beneficial if the preserved distances contribute to the application objective (i.e. contain information of label distribution), and oppose the learning problem otherwise.*

In the following, we analyze the practical scenario of class imbalance to showcase Observation 1.

### 3.2 DISTANCE PRESERVATION UNDER CLASS IMBALANCE

In practice, relevancy information of features is not available and popular options preserve differences between features blindly without regard of application (Miyato et al., 2018; Gulrajani et al., 2017). A common example where this characteristic is problematic is class imbalance. Here, information is either over- or under-represented in the training set, resulting in an increase of application-irrelevant data. In this subsection, we investigate the generalization performance when preserving features in the output under different severities of class imbalance. We find that distance constraints result in performance decline under high imbalance severities. In addition to generalization performance, we further investigate the uncertainty estimates under class imbalance in Appendix C.1.

**Experimental Setup** To illustrate class imbalance, we artificially imbalance the CIFAR100 benchmark by removing either training or test samples of a previously balanced class. The portion of classes we artificially im-

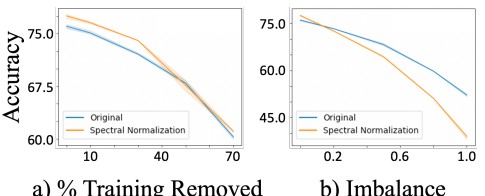

a) % Training Removed    b) Imbalance

Figure 2: Classification accuracy with and without distance preservation in the output: a) uniformly removed training and test data (left); b) class imbalance at different severity levels (right). In both graphs, we show the classification accuracy on the y-axis. The x-axis on the left graph represents the percentage of uniformly removed data, on the right the axis represents the fraction of imbalanced classes. The zero point on the x-axis is equivalent for both scenarios and represents the standard CIFAR100 benchmark without imbalance or data removal.

balance determines the severity of imbalance. For our experiments, we enforce distance preservation through spectral normalization in combination with residual connections (Miyato et al., 2018). We choose spectral normalization due to its simplicity and often stronger performance than the double sided gradient penalty (Gulrajani et al., 2017). We compare other distance-based methods in our benchmark experiments in Section 6. Full details on both imbalance method and experimental setup are provided in Appendix B.1.

**Accuracy Curves** We compare the classification accuracy with and without spectral normalization in Figure 2. In addition to class imbalance, we further consider settings where we randomly remove training samples (left graph). We show this setting to determine that the accuracy gain/loss from output feature preservation is dependent on the available information, not the number of samples. Our experiments highlight both advantages and disadvantages of preserving features in the output representation: if the target distribution is sufficiently similar to the input distribution, additional preserved features correlate with the generalization objective and results in a performance increase. This can be seen from the accuracy improvement with spectral normalization under low imbalance

severities or when samples are removed randomly. The opposite can be observed where, in contrast to random sample removal (left graph), we explicitly remove training and test samples to imbalance classes (right graph). Here, the training set contains more significant amounts of irrelevant information and spectral normalization significantly decreases the generalization performance.

## 4 OUR METHOD: TURING PROCESSES

### 4.1 TRANSITIONAL FEATURE PRESERVATION

Within the previous sections we found that feature preservation in the output can oppose the learning objective; an undesirable property for neural networks. To this end, a more prudent strategy involves preserving distance information in a collection of representations instead of a single output. Classic models that implement this property are ensembles (Lakshminarayanan et al., 2017). Uncertainty is encoded in the collection of ensemble models *without explicit preservation constraints* and the feature distance is then preserved in the difference or "spread" of the individual representations (Malinin et al., 2019). We formalize the concept within the context of distance preservation: given a set of neural network representations $\{h_{w_1}(\mathbf{x}), ..., h_{w_N}(\mathbf{x})\}$ and a transitional function $\Delta h : \mathcal{H}_1 \times \mathcal{H}_2 \times ... \times \mathcal{H}_N \to \mathcal{V}$, we seek to learn feature mappings that allow a distance $d_V$ within the transitional space that reflects the true distance $d_X$:

$$d_V(\Delta h(\mathbf{x}_1), \Delta h(\mathbf{x}_2)) = d_X(\mathbf{x}_1, \mathbf{x}_2) \tag{4}$$

We refer to methods along Equation 4 as *transitional feature preservation* or TFP in short. While powerful iterative methods such as ensembles implement TFP, their evaluation requires several forward passes and is frequently infeasible due to time or space limitations. In this paper, we implement TFP in single-pass uncertainty estimation by considering a linear combination of intermediate layer representations within the neural network. In particular, we find that the linear combination of intermediate distances $\sum_{l=0}^{L} r_l d_{H_l}(h_{w_l}(\mathbf{x}_1), h_{w_l}(\mathbf{x}_2))$ is distance preserving when the first layer is collapse resistant (i.e. $d_{H_0}(h_{w_0}(\mathbf{x}_1), h_{w_0}(\mathbf{x}_2)) \neq 0$) for $d_X(\mathbf{x}_1, \mathbf{x}_2) \neq 0$). This requirement is different from Equation 2 as it allows distance contraction or expansion of $d_{H_0}$ with respect to $d_X$, not full preservation. Proposition 1 makes the concept more precise.

**Proposition 1** (Transitional Feature Preservation in Intermediate Representations)**.** *Consider the neural network mapping* $h_w : \mathcal{X} \to \mathcal{H}$ *with the layered architecture* $h_w = h_{w_0} \circ h_{w_1} ... \circ h_{w_L}$, *where the first layer* $h_{w_0}$ *is collapse resistant with respect to the input space,* $d_{H_0}(h_{w_0}(\mathbf{x}_1), h_{w_0}(\mathbf{x}_2)) \neq 0$ *for* $d_X(\mathbf{x}_1, \mathbf{x}_2) \neq 0$. *Then there exists a* $C \in \mathbb{R}$ *such that*

$$\sum_{l=0}^{L} r_l d_{H_l}(h_{w_l}(\mathbf{x}_1), h_{w_l}(\mathbf{x}_2)) = C * d_X(\mathbf{x}_1, \mathbf{x}_2),$$

*where* $C = 1$ *under an appropriate choice of* $r_l$. *In other words, there exists a linear combination of intermediate representations that is feature preserving in transition - i.e. satisfies Equation 4.*

We provide proof of Proposition 1 in Appendix A.2. Note that Proposition 1 assumes collapse resistance in the first layer. In practice, this can be achieved by enforcing preservation constraints such as spectral normalization on the first layer only, which theoretically comes with the same risks outlined in Section 3 (to a significantly lesser degree). Empirically, we found that omitting the constraint entirely does not compromise the quality of the uncertainty estimates suggesting that the first layer rarely collapses data points in practice.

### 4.2 ALGORITHM

Our method consists of three components: a principal feed-forward network, shallow-deep network exits with individual internal classifiers, and a combination head (Figure 3). During training, the shallow-deep network exits are trained jointly with the feed-forward component, while the combination head is fitted after optimization on a validation set extracted from the training data $\mathcal{X}_{ID}$. We emphasize that our method *does not require out-of-distribution validation samples* as the combination head is fitted with data from the training set. During inference, the input sample traverses both the feed-forward network, as well as the shallow exits. The final output prediction is derived from

the main network, while the uncertainty score is derived from a combination of the intermediate output logits. In addition to our description in the main paper, we provide additional implementation details and algorithm pseudo code in Appendix D.

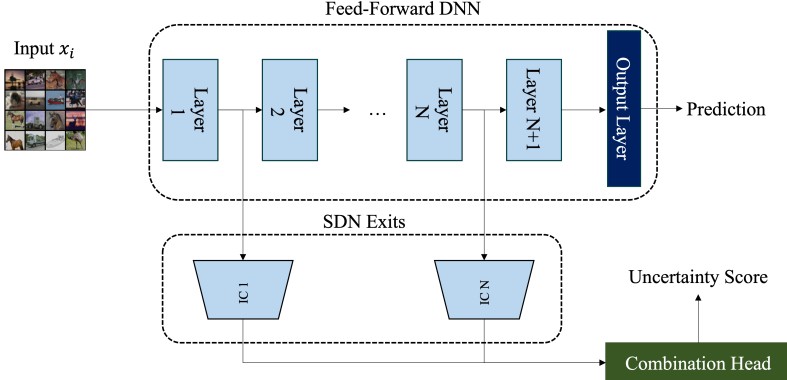

Figure 3: Workflow of our method during inference. The architecture consists of a main network, internal classifiers (IC), as well as a combination head. During inference, the input traverses the main network, as well as the internal classifiers. The prediction is obtained from the main network output, while the uncertainty score is obtained from a combination of internal classifier outputs.

**Shallow-Deep Network Exits**    Shallow-Deep networks (SDNs) (Kaya et al., 2019) were originally introduced in the context of computation reduction. A SDN is a modified version of a conventional DNN where additional internal classifiers are placed on intermediate representations to produce preliminary predictions. In our context, we utilize internal classifiers to produce intermediate logits for uncertainty estimation. Formally, given the intermediate layer $l$ of the principal feed-forward network with the latent representation $h^l_{w_l}(\mathbf{x}) \in \mathbb{R}^{M_l}$, an internal classifier is a shallow network $f^l_{w_l} : \mathbb{R}^{M_l} \to \mathbb{R}^K$ that produces prediction logits for $K$ classes. Regarding positioning, we place internal classifiers uniformly across the feed forward network, similar to Kaya et al. (2019).

**Training Procedure**    We train internal classifiers jointly with the feed-forward network, similar to Kaya et al. (2019). For this purpose, we propose a weighted loss function for each internal classifier, as well as the principal network output. Given the final logits of the principle network $h_w(\mathbf{x})$, an appropriate proper scoring rule (or loss) $L_s$, the SDN Loss is defined as

$$L_{SDN}(\mathbf{x}, y) = p_0 L_s(h_w(\mathbf{x}), y) + \sum_{i=1}^{N_{IC}} p_i L_s(f^i_{w_l}(\mathbf{x}), y). \tag{5}$$

where $p_i$ represent individual loss weights, and $N_{IC}$ is the total number of internal classifiers. In our experiments, we found that equal weighting (i.e. $p_k = \frac{1}{N_{IC}+1}$)) produces sufficient results but emphasize that other weighting schemes may be more desirable depending on the application.

**Combination Head**    While heavily utilizing early layer representations is intuitive, we found that relying on shallow outputs exclusively results in poor uncertainty scores. Specifically, in addition to increased domain awareness, early intermediate outputs contain significant amounts of application-irrelevant information that results in unreliable uncertainty estimates. For this purpose, we propose a fusion scheme by first calculating individual uncertainty scores and subsequently combining them by a weighted scaled sum.

$$u_{final}(\mathbf{x}) = \frac{1}{\sum_{i=1}^{N_{IC}} r_i} \sum_{i=1}^{N_{IC}} r_i u_s(f^i_{w_i}(\mathbf{x})). \tag{6}$$

In Equation 6, $u_s$ represents the individual uncertainty score, and $r_i$ the score weights. While principally any uncertainty score can be used, we replace the output layer of the internal classifiers with approximate gaussian processes and calculate uncertainty similar to (Liu et al., 2020). Our

Table 1: OOD detection and classification accuracy on the CIFAR10 dataset.

| Architecture | Algorithms | Preservation Constraint | Accuracy | CIFAR10-C | OOD AUROC CIFAR100-C | SVHN |
|---|---|---|---|---|---|---|
| ResNet-50 | DNN | No Constraint | **95.533 ± 0.080** | 0.715 ± 0.007 | 0.903 ± 0.000 | 0.927 ± 0.015 |
| | Energy-Based (Liu et al., 2021) | | | 0.688 ± 0.016 | 0.879 ± 0.003 | 0.919 ± 0.008 |
| | SNGP(Liu et al., 2020) | SN | 95.033 ± 0.076 | 0.721 ± 0.007 | 0.928 ± 0.005 | **0.976 ± 0.003** |
| | DUQ(Van Amersfoort et al., 2020) | GP | 88.867 ± 0.211 | 0.618 ± 0.003 | 0.824 ± 0.008 | 0.829 ± 0.016 |
| | TUrING Processes | No Constraint | 94.880 ± 0.324 | **0.738 ± 0.006** | **0.936 ± 0.003** | 0.946 ± 0.012 |
| ResNet-101 | DNN | No Constraint | **95.837 ± 0.103** | 0.690 ± 0.006 | 0.894 ± 0.001 | 0.922 ± 0.012 |
| | Energy-Based (Liu et al., 2021) | | | 0.715 ± 0.006 | 0.849 ± 0.009 | 0.890 ± 0.034 |
| | SNGP(Liu et al., 2020) | SN | 91.907 ± 0.183 | 0.636 ± 0.009 | 0.912 ± 0.015 | 0.906 ± 0.018 |
| | DUQ(Van Amersfoort et al., 2020) | GP | 89.427 ± 0.315 | 0.620 ± 0.003 | 0.833 ± 0.004 | 0.830 ± 0.004 |
| | TUrING Processes | No Constraint | 94.257 ± 0.349 | **0.722 ± 0.006** | **0.937 ± 0.003** | **0.938 ± 0.004** |
| ResNet-152 | DNN | No Constraint | **95.877 ± 0.097** | 0.690 ± 0.006 | 0.891 ± 0.000 | 0.928 ± 0.005 |
| | Energy-Based (Liu et al., 2021) | | | 0.665 ± 0.008 | 0.849 ± 0.004 | 0.905 ± 0.018 |
| | SNGP(Liu et al., 2020) | SN | 90.510 ± 0.814 | 0.636 ± 0.009 | 0.899 ± 0.016 | 0.847 ± 0.020 |
| | DUQ(Van Amersfoort et al., 2020) | GP | 91.263 ± 0.185 | 0.623 ± 0.002 | 0.731 ± 0.059 | 0.842 ± 0.012 |
| | TUrING Processes | No Constraint | 94.213 ± 0.777 | **0.722 ± 0.006** | **0.927 ± 0.004** | **0.945 ± 0.004** |

choice is based on simplicity and the strong performance of approximate Gaussian processes. We call our resulting method *Transitional Uncertainty with Intermediate Neural Gaussian Processes* or TUrING Processes in short.

**Fitting the Combination Head**   Fitting the weights $r_i$ in Equation 6 is more involved. While several methods exist to fit uncertainty parameters (Lee et al., 2023; 2018; Liang et al., 2017), they require either a) access to a small set of $\mathcal{X}_{OOD}$, and/or b) access to all labels in $\mathcal{X}_{ID}$. In this paper, we assume neither. Our fitting algorithm requires two steps. First, we derive proxy labels from the small validation set. Second, we derive the weight parameters by formulating a binary classification problem, using the individual uncertainty scores $u_s(f_{w_i}^i(\mathbf{x}))$, and proxy labels $s(\mathbf{x})$ derived from the disagreement in between different SDN exits. In Appendix C.2, we study the proxy labels in detail and compare disagreement in SDN exits with ensembles. Note, that our algorithm assumes that the validation set represents a small amount of *unlabeled* samples originating from $\mathcal{X}_{ID}$ and does *not* require out-of-distribution samples or additional data of any kind. We define our proxy labels through disagreement in the form of prediction switches between internal classifiers. Given a validation sample $\mathbf{x}_{val}$, we define a prediction switch as

$$s(\mathbf{x}_{val}) = min(\sum_{i=2}^{N_{IC}} \mathbf{1}_{f_{w_i}^i != f_{w_{i-1}}^{i-1}}, 1) \tag{7}$$

where $f_{w_i}^i$ are abbreviations for the internal classifier predictions $pred(f_{w_i}^i(\mathbf{x}_{val})$, and $\mathbf{1}_{f_{w_i}^i != f_{w_{i-1}}^{i-1}}$ is a binary variable reducing to one if two subsequent classifier predictions differ or zero otherwise. Our choice regarding the disagreement label is based on simplicity. By evaluating prediction switches, we reduce the tuning process to a binary classification problem allowing a partition of the validation set into coarse high-, and one low-uncertainty subgroups. Specifically, we classify the sample $\mathbf{x}_{val}$ as high-uncertainty if $s(\mathbf{x}_{val})$ amounts to one and as low-uncertainty otherwise. Subsequently, we derive the weighting parameters through logistic regression, where we map the individual uncertainty scores to the $u_s(f_{w_i}^i(\mathbf{x}_{val}))$ to the corresponding subgroup $s(\mathbf{x}_{val})$:

$$r_1, ..., r_{N_{IC}} = LR(\{s(\mathbf{x}_i), \mathbf{v}_i\}_{i=1}^{N_{val}}) \tag{8}$$

In our notation, $LR$ is an abbreviation for logistic regression, and $\mathbf{v}_i$ are the individual uncertainty scores $[u_s(f_{w_1}^1(\mathbf{x}_i)), ..., u_s(f_{w_{N_{IC}}}^{N_{IC}}(\mathbf{x}_i))]$ bundled into a single vector.

## 5 RELATED WORK

**Single-Pass Uncertainty Estimation**   Our work most closely relates with estimating uncertainty in a single forward pass. In this context, a significant amount of work includes replacing the loss function (Malinin & Gales, 2018; Hein et al., 2019; Sensoy et al., 2018), the output layer (Liu et al., 2021; Padhy et al., 2020; Bendale & Boult, 2016; Macêdo & Ludermir, 2022; Tagasovska & Lopez-Paz, 2019), or alternative gradient representations (Kwon et al., 2020; Prabhushankar & AlRegib; Lee et al., 2023). While several methods are promising, they do not explicitly consider information

Table 2: OOD detection and classification accuracy on the CIFAR100 dataset.

| Architecture | Algorithms | Preservation Constraint | Accuracy | CIFAR10-C | OOD AUROC CIFAR100-C | SVHN |
|---|---|---|---|---|---|---|
| ResNet-50 | DNN | No Constraint | 77.623 ± 0.276 | 0.832 ± 0.003 | 0.680 ± 0.004 | 0.859 ± 0.013 |
| | Energy-Based (Liu et al., 2021) | | | 0.829 ± 0.008 | 0.682 ± 0.006 | 0.815 ± 0.050 |
| | SNGP(Liu et al., 2020) | SN | 75.083 ± 0.889 | 0.821 ± 0.010 | 0.707 ± 0.005 | 0.900 ± 0.008 |
| | DUQ(Van Amersfoort et al., 2020) | GP | - | - | - | - |
| | TUrING Processes | No Constraint | **78.437 ± 0.223** | **0.868 ± 0.002** | **0.738 ± 0.002** | **0.955 ± 0.008** |
| ResNet-101 | DNN | No Constraint | 77.257 ± 0.285 | 0.834 ± 0.003 | 0.671 ± 0.005 | 0.851 ± 0.024 |
| | Energy-Based (Liu et al., 2021) | | | 0.836 ± 0.004 | 0.674 ± 0.007 | 0.856 ± 0.042 |
| | SNGP(Liu et al., 2020) | SN | 74.380 ± 1.978 | 0.816 ± 0.028 | 0.674 ± 0.031 | 0.906 ± 0.027 |
| | DUQ(Van Amersfoort et al., 2020) | GP | - | - | - | - |
| | TUrING Processes | No Constraint | **78.550 ± 0.213** | **0.863 ± 0.004** | **0.730 ± 0.002** | **0.959 ± 0.001** |
| ResNet-152 | DNN | No Constraint | 78.160 ± 0.242 | 0.830 ± 0.002 | 0.674 ± 0.002 | 0.851 ± 0.010 |
| | Energy-Based (Liu et al., 2021) | | | 0.828 ± 0.002 | 0.674 ± 0.002 | 0.852 ± 0.009 |
| | SNGP(Liu et al., 2020) | SN | 74.077 ± 2.631 | 0.822 ± 0.028 | 0.659 ± 0.039 | 0.889 ± 0.015 |
| | DUQ(Van Amersfoort et al., 2020) | GP | - | - | - | - |
| | TUrING Processes | No Constraint | **78.877 ± 0.311** | **0.857 ± 0.010** | **0.715 ± 0.010** | **0.926 ± 0.031** |

preservation within the representations of the network. In contrast, recent methods consider feature preservation within the output as a vital component for reliable uncertainty scores (Liu et al., 2020; Van Amersfoort et al., 2020; van Amersfoort et al., 2021; Mukhoti et al., 2023). Our work complements these approaches by considering feature preservation with intermediate representations without explicit constraints. Finally, several approaches explore intermediate representations for the application of uncertainty estimation (Lee et al., 2023; 2018; Guo et al., 2017; Liang et al., 2017). However, they assume access to out-of-distribution validation samples and/or a fully labeled training set. Our work is complementary by investigating feature preservation without access to a out-of-distribution validation set and does not assume that all samples are labeled in training.

**Iterative Uncertainty Estimation** In addition to single-pass uncertainty, significant related work exists in iterative uncertainty estimation. With iterative uncertainty estimation, we refer to methods requiring several forward passes for computation. Here, the state-of-the-art are deep ensembles (Lakshminarayanan et al., 2017), as well as several parameter-efficient counterparts (Wen et al., 2020; Dusenberry et al., 2020; Thiagarajan et al., 2022). Further examples include Bayesian Neural Networks (Wenzel et al., 2020; Osawa et al., 2019) and MC-Dropout (Gal & Ghahramani, 2016). In practice, these methods tend to render powerful uncertainty estimates but require several forward passes to compute, limiting their applicability in practice. In Appendix E, we provide additional related work on distance preserving neural networks.

## 6 BENCHMARK EXPERIMENTS

### 6.1 CIFAR10 AND CIFAR100

We start by demonstrating our uncertainty method on standardized benchmarks in out-of-distribution (OOD) detection. The following combinations are evaluated: CIFAR10 vs. CIFAR10-C/CIFAR100-C/SVHN and CIFAR100 vs. CIFAR10-C/CIFAR100-C/SVHN (Krizhevsky et al., 2009; Netzer et al., 2011; Hendrycks & Dietterich, 2019). In addition to a standard softmax DNN, we compare against three single-pass uncertainty baselines that do not require additional OOD data: the energy-based model (Liu et al., 2021), DUQ (Van Amersfoort et al., 2020), and SNGP (Liu et al., 2020). We choose these three methods because 1) their strong empirical performance and 2) they utilize three popular methods for feature preservation in the network output. DUQ preserves features with a double sided gradient penalty (GP) (Gulrajani et al., 2017), SNGP implements spectral normalization (SN) (Miyato et al., 2018), and the energy based model relies on the softmax density without regularization (No constraint). To investigate robustness with respect to network complexity, we consider three architectures with residual connections and varying depth: ResNet architectures (He et al., 2016) with 50, 101, and 152 layers. We restrict our experiments to these architectures as SNGP requires residual connections for its full functionality. For fine-tuning the uncertainty weights, we partition a small amount (10%) of training samples and remove the labels to perform the unsupervised fitting algorithm. In Table 1 and Table 2, we report the AUROC scores for training on CIFAR10 and CIFAR100 respectively. When evaluating the corruption datasets, CIFAR10-C and CIFAR100-C, we average all corruption types and intensities. Further details on implementation and feature preservation can be found in Appendix B.3 and B.2. In addition, we investigate calibration, runtime, and imbalanced settings in Appendix C.3, and C.4.

Table 3: AUROC and classification accuracy on on the organ{C, A, S} datasets.

| Dataset | Algorithms | Preservation Constraint | Accuracy | AUROC organA | AUROC organC | AUROC organS |
|---|---|---|---|---|---|---|
| organA | DNN | No Constraint | 94.602 ± 0.388 | - | 0.906 ± 0.004 | 0.850 ± 0.007 |
| | Energy-Based (Liu et al. 2021) | | | - | 0.887 ± 0.005 | 0.841 ± 0.008 |
| | SNGP(Liu et al. 2020) | SN | 93.906 ± 0.297 | - | **0.907 ± 0.010** | 0.857 ± 0.006 |
| | TUrING Processes | No Constraint | **95.254 ± 0.191** | - | **0.915 ± 0.003** | **0.869 ± 0.003** |
| organC | DNN | No Constraint | **92.106 ± 0.176** | 0.884 ± 0.001 | - | 0.780 ± 0.005 |
| | Energy-Based (Liu et al. 2021) | | | 0.857 ± 0.001 | - | 0.751 ± 0.006 |
| | SNGP(Liu et al. 2020) | SN | 90.941 ± 0.530 | 0.849 ± 0.008 | - | 0.765 ± 0.007 |
| | TUrING Processes | No Constraint | **92.122 ± 0.227** | **0.894 ± 0.003** | - | **0.794 ± 0.003** |
| organS | DNN | No Constraint | **80.258 ± 0.299** | 0.754 ± 0.010 | 0.815 ± 0.003 | - |
| | Energy-Based (Liu et al. 2021) | | | 0.733 ± 0.012 | 0.775 ± 0.005 | - |
| | SNGP(Liu et al. 2020) | SN | 79.918 ± 0.280 | 0.707 ± 0.009 | 0.790 ± 0.010 | - |
| | TUrING Processes | No Constraint | **80.002 ± 0.126** | **0.778 ± 0.001** | **0.822 ± 0.003** | - |

Our method outperforms the other single-pass methods despite having access to less training annotations. This holds true over varying architectures and training datasets. In particular, methods trained with spectral normalization achieve lower accuracy with increasing network depth. For instance, the accuracy of SNGP reduces nearly three percent when extending ResNet-50 to ResNet-101. We relate this observation to the scaling of the Lipschitz bounds. Specifically, the lower and upper bounds scale exponentially with the number of layers and are tighter for shallow models while looser for deeper ones. Hence, the preservation constraint is weaker for deeper models and fails to maintain the distance between data-points. The remaining methods deploy a different feature preservation strategy and are thus agnostic to this effect. Further, DUQ did not converge on CIFAR100 due to training instabilities. These arise when the class centroids get noisy from increasing class and data complexity.

## 6.2 MEDICAL MODALITIES

We consider a natural example where the training data contains different information as the test set. For this purpose, we benchmark TUrING Processes on three CT scan datasets from Yang et al. (2023). All three datasets contain CT scans of the same eleven body organs and are named after the three planes (axial, coronal, and saggittal) in which the data was collected. In our experiments, we train on one plane and perform misclassification detection on the combined test set of the original plane and an additional plane. We report AUROC and accuracy on the in-domain test set in Table 3. Each row shows the a different training set, while each column refers to the test set that is combined with the in-distribution test set. All experiments are performed with a ResNet-50 architecture and we use the same hyperparameters as in our previous experiments. TUrING Processes match or outperform competing methods. Similar to our previous observations, we note that SNGP does not perform well when the information within the training set does not correlate well with the test set and supports our usage of TFP in TUrING Processes.

## 7 DISCUSSION AND LIMITATIONS

A central observation we made in this work is that *enforcing feature preservation by constraining model representations can be harmful to the model performance* and highlight application relevance as a key requirement for effective representation constraints. For practical applications of uncertainty estimation, the characteristic is undesirable as training distributions can severely differ from deployment. We propose single-pass transitional feature preservation through intermediate representations to address these disadvantages. While our approach is effective and simple, we do not claim that the illustrated improvements solve the problem of feature preservation in single-pass uncertainty estimation entirely. In particular, we propose one instance of TFP through SDNs that comes with its own set of limitations: similar to iterative methods, the success depends on the amount of source representations in $\Delta h$ to preserve features. For this purpose, SDNs are less effective on shallow architectures with fewer intermediate options to extract from. Finally, we chose the combination of SDNs with Gaussian Processes out of simplicity and the strong empirical performance. However, key novelties of this paper (SDNs and singe-pass TFP) are not limited to one uncertainty score (Gaussian Processes) and can be viewed as building blocks for single-pass uncertainty methods. We encourage researchers to implement different combinations of single-pass TFP and either existing or novel uncertainty scores to advance the critical field of single-pass uncertainty estimation.

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
