# A    PROOFS

## A.1    MUTUAL INFORMATION OF DISTANCES

In this section, we discuss connecting the learning problem in Equation 1 with distances in the feature plane. To establish a dependency between Equation 1 and feature distances, we first define sets where the distance $d_X$ is bijective. Mathematically, this is equivalent to restricting sets to unique distance values; sets we define as unique distance sets. Definition 1 formalizes the concept.

**Definition 1** (Unique Distance Set and Partition). *Consider the metric space $(\mathcal{X}, d_X)$ with a corresponding random variable $X \sim p_X$ describing the input distribution with density $p_X$. We define a unique distance set $\mathcal{M} \subset \mathcal{X}$ as a set in $\mathcal{X}$ possessing unique distances with respect to $d_X$ and an arbitrary but fixed anchor point $\mathbf{x}_a \in \mathcal{X}$.*

$$\mathcal{M} = \{\mathbf{x}_i, \mathbf{x}_j, \mathbf{x}_a \in \mathcal{X} : d_X(\mathbf{x}_a, \mathbf{x}_j) \neq d_X(\mathbf{x}_a, \mathbf{x}_i),\ i \neq j \neq a\}$$

*Further, we define a partition $\mathcal{X} = \bigcup_{k \in [1, N_p]} \mathcal{M}^k : \bigcap_{k \in [1, N_p]} \mathcal{M}^k = \emptyset$ over unique distance sets $\mathcal{M}^k$ as a unique partition set. Equivalently, we define $M \sim p_M$ and $M^k \sim p_{M^k}$ as the corresponding random variables with their respective probability densities.*

With the help of unique distance sets, we can formulate the proof for Equation 3.

*Proof.* We first formulate Equation 1 as an optimization objective over unique distance sets. Since the optimization over each individual set can be viewed as a separate learning task, we can rewrite the objective as a summary of the mutual information over individual distance sets:

$$h^*(X) = \underset{h_w : \{I(h_w(M^k); Y^k) = I(M^k; Y^k),\ k \in [1, N_p]\}}{\arg\min} \sum_k I(h_w(M^k); M^k) \qquad (9)$$

Given the anchor point $\mathbf{x}_k$ for $\mathcal{M}^k$, $f_X(\mathbf{x}) = d_X(\mathbf{x}_k; \mathbf{x})$ represents an injection within the individual subset $\mathcal{M}^k$. The characteristic is relevant, as we utilize the transformation invariance property of the mutual information. Assuming $h_w$ preserves the unique distance property according to Equation 2, we rewrite Equation 9 as

$$
\begin{aligned}
h^*(X) &= \underset{h_w : \{I(h_w(M^k); Y^k) = I(M^k; Y^k),\ k \in [1, N]\}}{\arg\min} \sum_k I(h_w(M^k); M^k) \\
&= \underset{h_w : \{I(d_H(h_w(\mathbf{x}_k); h_w(M^k)); Y^k) = I(M^k; Y^k),\ k \in [1, N]\}}{\arg\min} \sum_k I(d_H(h_w(\mathbf{x}_k); h_w(M^k)); d_X(\mathbf{x}_k; M^k)) \\
&= \underset{h_w : \{I(f_H^k(h_w(M^k)); Y^k) = I(M^k; Y^k),\ k \in [1, N_p]\}}{\arg\min} \sum_k I(f_H^k(h_w(M^k)); f_X^k(M^k)).
\end{aligned}
$$

$$(10)$$

$\square$

## A.2    TRANSITIONAL FEATURE PRESERVATION OF INTERMEDIATE REPRESENTATIONS

In this section, we discuss the proof of Proposition 1.

*Proof.* To prove Proposition 1, we utilize concepts of metric distortion from metric embedding theory (Abraham et al., 2006; Chennuru Vankadara & von Luxburg, 2018). Specifically, neural network layers can be characterized by the distortion they introduce to the input space. We define the network distortion coefficient of a given layer $l$ as

$$\rho_l(\mathbf{x}_1, \mathbf{x}_2) = \begin{cases} \frac{d_{H_l}(h_{w_l}(\mathbf{x}_1), h_{w_l}(\mathbf{x}_2))}{d_{H_{l-1}}(h_{w_{l-1}}(\mathbf{x}_1), h_{w_{l-1}}(\mathbf{x}_2))}, & \text{if } d_{H_{l-1}}(h_{w_{l-1}}(\mathbf{x}_1), h_{w_{l-1}}(\mathbf{x}_2)) \neq 0 \\ 1, & \text{otherwise} \end{cases}. \qquad (11)$$

The interpretation of Equation 11 is simple. If the previous layer does not collapse the input distances $(d_{H_{l-1}}(h_{w_{l-1}}(\mathbf{x}_1), h_{w_{l-1}}(\mathbf{x}_2)) \neq 0)$, the ratio between both distances characterizes the distance distortion of layer $l$. Specifically, we have *distance contraction* if $\rho_l(\mathbf{x}_1, \mathbf{x}_2)$ is less than one. Here, zero represents the corner case when the layer collapses the input to a single point $(d_{H_l}(h_{w_l}(\mathbf{x}_1), h_{w_l}(\mathbf{x}_2)) = 0)$. Further, we have *distance expansion* when $\rho_l(\mathbf{x}_1, \mathbf{x}_2)$ is greater than one, i.e. when the layer increases the distance between the points. Finally, we have perfect distance preservation in the case of $\rho_l(\mathbf{x}_1, \mathbf{x}_2) = 1$. When the previous layer collapses the input points, the layer $l$ receives the same input for $\mathbf{x}_1$ and $\mathbf{x}_2$, resulting a perfect distance preservation as the distance between the same point is trivially zero.

The following relationship between distortion and network distances is important.

$$d_{H_l}(h_{w_l}(\mathbf{x}_1), h_{w_l}(\mathbf{x}_2)) = \rho_l(\mathbf{x}_1, \mathbf{x}_2) * d_{H_{l-1}}(h_{w_{l-1}}(\mathbf{x}_1), h_{w_{l-1}}(\mathbf{x}_2)) \tag{12}$$

Equation 12 follows directly from the the fact that the input of a given layer $l$ is the output of the previous layer $h_w = h_{w_0} \circ h_{w_1} ... \circ h_{w_L}$. When the previous layer does not collapse the input Equation 12 directly follows from the definition of $\rho_l$. In the case of feature collapse within the previous layer $(d_{H_{l-1}}(h_{w_{l-1}}(\mathbf{x}_1), h_{w_{l-1}}(\mathbf{x}_2)) = 0)$, the input to the next layer is the collapsed point and $d_{H_l}(h_{w_l}(\mathbf{x}_1), h_{w_l}(\mathbf{x}_2)) = 0$ satisfying Equation 12.

Using the network distortion coefficient we can rewrite the linear combination of distances as a function of the input distances $d_X$:

$$
\begin{aligned}
d_{SDN}(\Delta h(\mathbf{x}_1), \Delta h(\mathbf{x}_2)) &= \sum_{l=0}^{L} r_l d_{H_l}(h_{w_l}(\mathbf{x}_1), h_{w_l}(\mathbf{x}_2)), \\
&= \sum_{l=0}^{L} r_l * d_X(\mathbf{x}_1, \mathbf{x}_2) * \prod_{i=0}^{l} \rho_i(\mathbf{x}_1, \mathbf{x}_2), \\
&= d_X(\mathbf{x}_1, \mathbf{x}_2) * \sum_{l=0}^{L} r_l * \prod_{i=0}^{l} \rho_i(\mathbf{x}_1, \mathbf{x}_2), \\
&= d_X(\mathbf{x}_1, \mathbf{x}_2) * C.
\end{aligned}
\tag{13}
$$

The first derivation follows from a recursive application of Equation 12. The second, from the independence of $d_X(\mathbf{x}_1, \mathbf{x}_2)$ from both $i$ and $l$.

We note that Equation 13 satisfies Equation 4 when an appropriate weight choice $r_l$ results in $C = 1$. A solution for $r_l$ only exists when the first layer is collapse resistant, i.e. when $\rho_0(\mathbf{x}_1, \mathbf{x}_2) \neq 0$ for $d_X(\mathbf{x}_1, \mathbf{x}_2) \neq 0$; a requirement for Proposition 1. $\qquad\square$

### A.3 Mutual Information of Intermediate Representations

In this section, we discuss how intermediate representations aid in increasing the information of the full input distribution within the uncertainty source representation. As discussed in Section 2.1, effective uncertainty estimation is contingent on modelling information of the full input space $\mathcal{X}$ (not just $\mathcal{X}_{ID}$) to differentiate the training distribution from the test distribution. Within the context of the mutual information in neural networks (Tishby et al., 2000), achieving this is equivalent to maintaining the mutual information between the uncertainty source representation $Z$ (i.e. the representation used to compute the uncertainty $u(Z)$), and the input $X$.

$$I(Z; X). \tag{14}$$

A differentiator for uncertainty estimators is therefore their uncertainty source $Z$. We show that combining intermediate layers has favorable uncertainty properties in comparison to a conventional neural network output - i.e. our method maintains the mutual information $I(Z; X)$ more effectively.

In our algorithm, we measure uncertainty from a combination of intermediate layers $h_{w_l}^l$ instead of the final output. Within the context of mutual information, the joint representation $Z = h_{w_1}^1, ..., h_{w_L}^L$ maintains the following relationship for a layered neural network:

$$I(h_{w_1}^1, ..., h_{w_L}^L; X) \geq I(h_w(X); X) \tag{15}$$

The interpretation of Equation 15 is simple. Our method preserves information by extracting features before they are collapsed by subsequent network components. Hence, the mutual information with respect to the input is larger when intermediate representations are utilized in comparison to the final output exclusively. We further provide proof for Equation 15:

*Proof.* For our discussion, we utilize data processing inequality (Cover & Thomas, 2006) within the context of neural networks. Specifically, given an intermediate layer $h_{w_l}^l$ the following relationship holds for any subsequent layers

$$I(h_{w_l}^l(X); X) \geq I(h_{w_{l+1}}^{l+1}(X); X). \tag{16}$$

The mutual information of the joint variable $Z = h_{w_1}^1, ..., h_{w_L}^L$ and input $X$ can be expressed with the chain rule of mutual information

$$\begin{aligned} I(Z; X) &= I(h_{w_1}^1(X); X) - I(h_{w_2}^2(X), ..., h_{w_L}^L(X); X | h_{w_1}^1(X)) \\ &= I(h_{w_1}^1(X); X) - H[X] + H[X | h_{w_2}^2(X), ..., h_{w_L}^L(X)] \\ &= I(h_{w_1}^1(X); X) \\ &\geq I(h_w(X); X) \end{aligned} \tag{17}$$

The first derivation comes from the chain rule of mutual information, the seconde from the definition of mutual information, and the third from the fact that layered neural networks form a Markov chain with $X \rightarrow h_{w_1}^1(X) \rightarrow ... \rightarrow h_{w_L}^L(X)$ (Tishby et al., 2000). The final inequality is a direct manifestation of Equation 16.

□

## B  IMPELEMENTATION DETAILS

In this appendix, we provide details of the different experimental setups and comparison methods used in this paper. All experiments are implemented with pytorch. When a implementation was publicly available, we heavily relied on it in our own code. This is the case for DUQ (https://github.com/y0ast/deterministic-uncertainty-quantification), and SNGP (https://github.com/google/uncertainty-baselines/blob/master/baselines/imagenet/sngp.py, as well as https://github.com/y0ast/DUE).

### B.1  SURFACE PLOTS AND CLASS DISTRIBUTION EXPERIMENTS

**Hyperparameter and Architecture Details**   In all experiments, we train a resnet-18 architecture (He et al., 2016) over 200 epochs and optimize with stochastic gradient descent with a learning rate of 0.01. We further decrease the learning rate by a factor of 0.2 in epochs 100, 125, 150, and 175 respectively, and use the data augmentations random crop, random horizontal flip, and cutout to increase the generalization performance. For our experiments, we deploy direct spectral normalization of the convolutional, and batch normalization layers to implement representational feature preservation. On the full CIFAR100 dataset, we achieve an overall classification accuracy of 77.41 % and 75.93 % for the model with and without spectral normalization respectively. We average our results over three random seeds.

**Imbalanced CIFAR100** We imbalance the dataset as follows: for a certain subset of classes $A \subset \{1, ..., K\}$, we reduce the the number of training samples by 80% and do not change the number of test samples. For a second subset $B \subset \{1, ..., K\}, A \cap B = \emptyset$, we reduce the number of test samples by 90% and do not change the number of training samples. As a result, the first subset of classes contains few training samples and a large amount of test samples, while the other set suffers from the opposite problem. The imbalance severity can be adjusted by the number of classes in both imbalanced subsets $A \cup B \subset \{1, ..., K\}$. For simplicity, we keep the same amount of classes in both subsets $A$ and $B$. In Figure 4, we show an overview of class distributions at different severity levels. Here, the top and bottom row contain the training and test distribution respectively. We perform our entire analysis on the CIFAR100 dataset as it represents a challenging benchmark with a large class variety. In addition, the dataset is fully balanced and contains the same class distribution for training and testing.

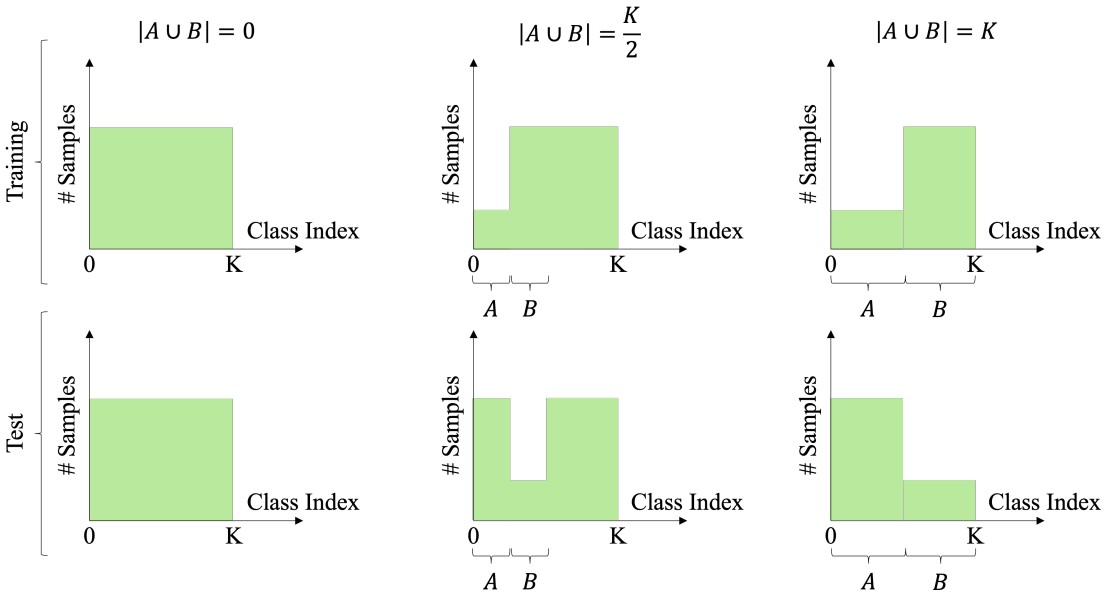

Figure 4: Toy example of class distributions at different imbalance severity levels. Each column represents a different severity level, and each column the training and test set distribution respectively.

## B.2 COMPARISON METHOD DETAILS

In the following, we provide details on each feature preservation method.

- *Energy-Based Model* (Liu et al., 2021): for the energy based model, uncertainty estimates are derived by replacing the final softmax layer with the unnormalized softmax density. No additional feature preservation constraint is used during training. In our experiments, we only compare against the version that does not require additional out-of-distribution data.

- *DUQ* (Van Amersfoort et al., 2020): in DUQ, uncertainty is inferred by the closest kernel distance. To ensure the preservation of features, DUQ implements the double sided gradient penalty, penalizing the squared distance of the gradient from a fixed value at every input point. In contrast to spectral normalization, it implements feature preservation with a *local* constraint, without explicit guarantees for data points outside of $\mathcal{X}_{ID}$.

- *SNGP* (Liu et al., 2020): SNGP infers uncertainty through distance awareness within the output. In the original paper, this is achieved by replacing the final output layer with a gaussian process approximation, and implementing spectral normalization in combination with residual layers. Direct spectral normalization on the weights provides a upper lipschitz bound while the combination with residual connections further ensures a lower lipschitz bound on the distance between two input points. Both bounds jointly result in feature preservation, as distances in between input points are approximately preserved when

traversing the network. In comparison to the gradient penalty, spectral normalization enforces a *global* constraint.

## B.3 OUT-OF-DISTRIBUTION EXPERIMENTS ON CIFAR10/100

In our out-of-distribution experiments, we use the same backbone residual architectures (ResNet-50, -101, and -152) with a batch size of 128. For all setups, we use the standard data augmentations random horizontal flip, random crop, and cutout. In the following, we describe the details for each method. All Results are averaged over three random seeds.

**Softmax DNN and Energy-Based Model**  We train both models with the SGD optimizer and an initial learning rate of 0.01. We optimize the model for 200 epochs and reduce the learning rate by a factor of 0.2 in epochs 100 and 150. For the energy-based model, we use the unnormalized softmax density, similar to other implementations (Mukhoti et al., 2023).

**DUQ**  Our DUQ models are trained with the SGD optimizer and a learning rate of 0.05. We train for 600 epochs and reduce the learning rate by factor 0.2 in epochs [300, 375, 450, 525]. For the gradient penalty weight, we perform the experiment with hyperparameters from the original paper (Van Amersfoort et al., 2020), as well as a newer implementation from Postels et al. (2022), and report the constellation with the highest accuracy value.

**SNGP**  We trained SNGP with the SGD optimizer and an initial learning rate of 0.01. We reduce the learning rate by a factor of 0.2 in epochs 100 and 150, and train for 200 epochs. We further use a spectral normalization coefficient of three.

**Our Method**  We train each SDN model with the SGD optimizer using an initial learning rate of 0.01. Further, we optimize the architecture for 400 epochs and reduce the learning rate by a factor of 0.2 in epoch 200, and 300. The architecture of the internal classifiers is similar to Kaya et al. (2019), with a single linear layer combined with a mixture of average-/max-pooling where necessary. The output layer is then fed into a GP layer, which has the same architecture as SNGP. We distribute the internal classifiers equally distanced across the network by placing a internal classifier on top of every third residual block. For ResNet-50 this is equivalent to every sixth layer, and every ninth layer for the remaining larger models. Our selection is geared towards simplicity and performance may be further improved with other uncertainty scores such as entropy or energy functions. We train each model with a equally weighted SDN loss.

## B.4 ARTIFICIAL DATASET

For our experiments in Section C.2, we train our models on an artificial spiral dataset with three different classes. Here, each spiral arm represents a class that starts in the center, and spirals for one full loop of 360°. In our ensemble experiments, we use a three-layer MLP architecture in an ensemble of ten. During training, we use a SGD optimizer with a learning rate of 0.008 and train each ensemble element for 400 epochs. To measure disagreement in between layers, we adjust the MLP architecture into a SDN, by placing an internal classifier on the first, and second layer respectively. We train the model with an SDN loss as described in Equation 5, optimize with the adam variant of SGD, and select a learning rate of 0.001. Due to the higher loss complexity, we train the SDN model for 800 epochs. To further measure disagreement, we utilize the same measure as previous works Mukhoti et al. (2023); Malinin et al. (2019).

$$disagreement(\mathbf{x}) = H[\frac{1}{N_{IC/E}} \sum_{i=1}^{N_{IC/E}} p(y|\mathbf{x}, w_i)] - \frac{1}{N_{IC/E}} \sum_{i=1}^{N_{IC/E}} H[p(y|\mathbf{x}, w_i)]. \tag{18}$$

Here, $N_{IC/E}$ denotes the number of internal classifiers or ensembles respectively, and $p(y|\mathbf{x}, w_i)$ the target posterior distribution of the individual model elements.

## C ADDITIONAL EXPERIMENTS

### C.1 SURFACE PLOTS FOR DISTANCE PRESERVATION UNDER CLASS IMBALANCE

In addition to class accuracy, we further wish to analyze uncertainty estimates under class imbalance. For this purpose, we plot both accuracy and the number of samples (sample concentration) with respect to imbalance severity and uncertainty scores (Figure 5). The accuracy plots provide information of the calibration capabilities in relation to class imbalance. Ideally, the uncertainty fully informs of the accuracy of a sample and the dependency is linear on the y-/z-plane (Guo et al., 2017). We note that a conventional neural network is not calibrated and overconfident in its prediction - the dependency is not linear. Spectral normalization significantly improves along this characteristic (top right plot), and improves linearity regardless of the imbalance severity. The bottom row complements our accuracy curves. For low imbalance severities the majority of samples concentrate low uncertainty/high accuracy regions on the x-/y-plane. However, the dependency inverts with increasing imbalance. Samples concentrate in high-uncertainty/low-accuracy regions complementing the accuracy decline in Figure 2.

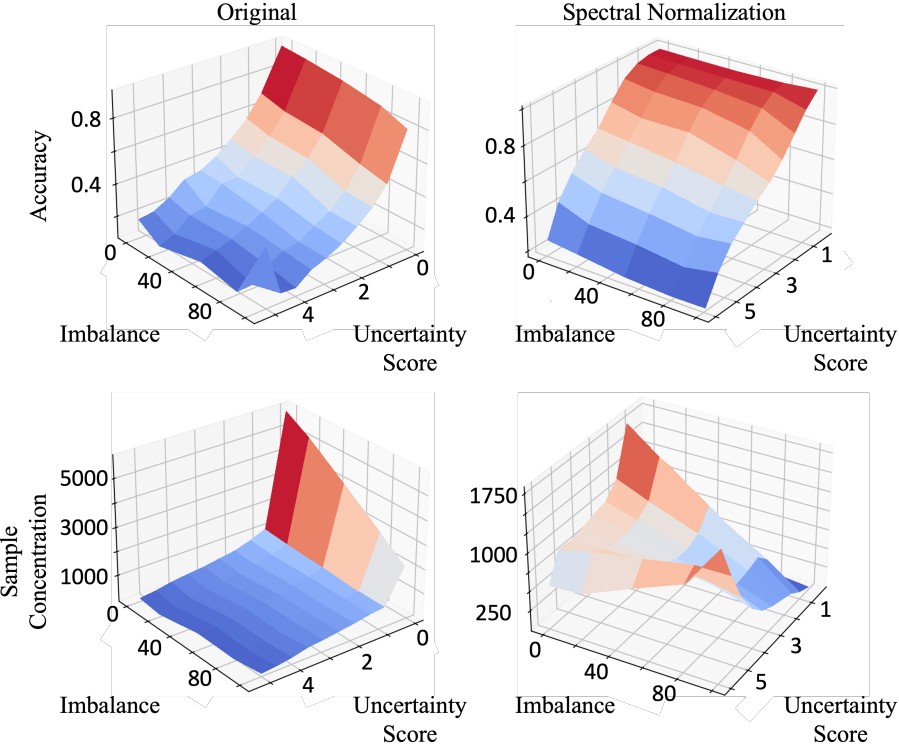

Figure 5: Accuracy and number of samples (sample concentration) with respect to dataset imbalance and average uncertainty score. In the the top row we show accuracy, in the bottom row we show sample concentration. The left column represents a conventional DNN while the right shows feature distance preservation through spectral normalization.

### C.2 DISAGREEMENT ANALYSIS OF PROXY LABELS

In this section, we provide a detailed analysis on the uncertainty proxy scores used in `TUrING Processes`. In our analysis, we compare the properties of intermediate representation with ensembles uncertainty scores and analyze the disagreement among both methods. Within the context of ensembles, disagreement is used to derive the difference or "spread" of ensembles and is frequently used directly as an uncertainty score in several contexts (Malinin et al., 2019). To showcase disagreement within intermediate representations, we compare against ensembles on an artificial

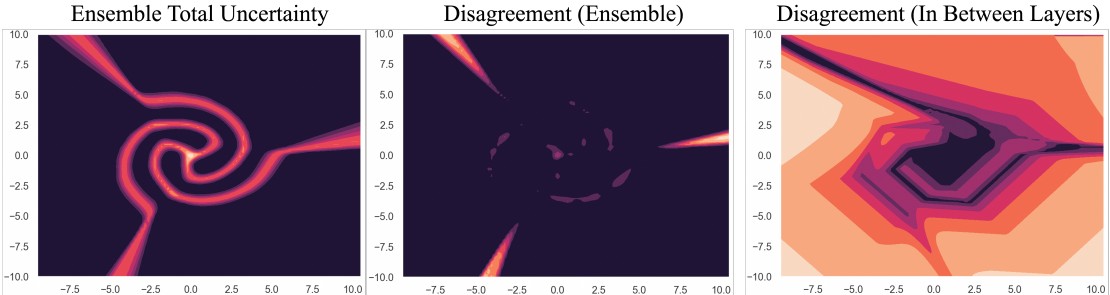

Figure 6: Comparison of disagreement in ensembles and intermediate representations trained on an artificial spiral dataset. The left image represents the total uncertainty of an ensemble of ten. In the center, we display model disagreement within the ensembles. The right depicts the disagreement within the intermediate representations of a single deterministic neural network. Dark red, or black represents low uncertainty while lighter shades of red or orange depict the opposite.

spiral dataset (Figure 6). The left image shows the total uncertainty of an ensemble of ten and is among the most common usages of ensembles. The center shows disagreement among the different ensemble models as derived by previous work (Malinin et al., 2019), and approximates uncertainty occurring due to data scarcity Kendall & Gal (2017). Ideally, the measure is low where sufficient data samples are available (center of the spiral), and increases where little or no data is available. The right shows the same measure of disagreement, with the exception of measuring in between individual SDN outputs instead of ensemble models. We note, that ensembles exhibit high disagreement near the decision boundaries exclusively, while the SDN model comprehensively approximates data scarcity in between layers. Our observations can be explained along the intuition of feature preservation. Ensembles measure disagreement among the output of the entire network architectures, and collapse important information from the input distribution that can be leveraged for uncertainty estimation. In contrast, intermediate representations contain more information of the input distribution and provide a coarse measure of data scarcity. Overall, we gather that disagreement in between internal classifiers is a reasonable approach to determine the hyperparameters $r_i$. Experimental details to produce Figure 6 can be found in Appendix B.4.

### C.3   CALIBRATION AND RUNTIME

We further investigate calibration as characteristic for uncertainty score quality. With calibration, we refer to the capability of the output score to be reflective of the actual generalization performance (Guo et al., 2017). For this purpose, we consider the expected calibration error (ECE) (Naeini et al., 2015) to measure miscalibration and show our results in Figure 4. Our setup is equivalent to our experiments on out-of-distribution detection and we further show the runtime normalized by the latency of a conventional DNN. Specifically, we measure the latency of a single batch for each model and divide by the latency of a conventional DNN. For all of our experiments we use a single NVIDIA GeForce GTX 1080 Ti. We note, that all algorithms are equivalent in terms of runtime and that our method matches or outperforms comparable single-pass methods in the majority of benchmarked constellations. Our results show both the runtime benefits and a high uncertainty estimation quality for our method.

### C.4   IMBALANCED OUT-OF-DISTRIBUTION EXPERIMENTS

In addition to standard out-of-distribution detection, we consider less informative training sets in the form of class imbalance. For this purpose, we unbalance the full 100 classes of the CIFAR100 dataset, as described in our previous analysis in Section 3. For the first 50 classes we reduce the training samples by 80% (400 samples) and maintain the same test set. For the remaining 50 classes, we reduce the test set by 90% (90 samples) and maintain all training samples. We use the same implementations as our previous out-of-distribution experiments and consider the three residual backbones ResNet-50, -101, and -152. We show the AUROC scores in Table 5. Complementary to our previous results, our method outperforms other single-pass uncertainty estimators despite having access

Table 4: Expected-Calibration-Error and Runtime on CIFAR10 and CIFAR100.

| Architecture | Algorithms | Runtime | ECE CIFAR10 | CIFAR100 |
|---|---|---|---|---|
| ResNet-50 | DNN Energy-Based (Liu et al., 2021) | 1x | $0.065 \pm 0.001$ | $1.110 \pm 0.002$ |
| | SNGP(Liu et al., 2020) | 1x | $\mathbf{0.010 \pm 0.002}$ | $0.720 \pm 0.002$ |
| | DUQ(Van Amersfoort et al., 2020) | 1x | $0.973 \pm 0.003$ | - |
| | TUrING Processes | 1x | $0.030 \pm 0.005$ | $\mathbf{0.658 \pm 0.004}$ |
| ResNet-101 | DNN Energy-Based (Liu et al., 2021) | 1x | $0.056 \pm 0.001$ | $1.108 \pm 0.011$ |
| | SNGP(Liu et al., 2020) | 1x | $0.039 \pm 0.013$ | $\mathbf{0.671 \pm 0.021}$ |
| | DUQ(Van Amersfoort et al., 2020) | 1x | $0.973 \pm 0.001$ | - |
| | TUrING Processes | 1x | $\mathbf{0.025 \pm 0.021}$ | $\mathbf{0.672 \pm 0.031}$ |
| ResNet-152 | DNN Energy-Based (Liu et al., 2021) | 1x | $0.055 \pm 0.001$ | $1.126 \pm 0.004$ |
| | SNGP(Liu et al., 2020) | 1x | $0.050 \pm 0.008$ | $\mathbf{0.698 \pm 0.005}$ |
| | DUQ(Van Amersfoort et al., 2020) | 1x | $0.623 \pm 0.002$ | - |
| | TUrING Processes | 1x | $\mathbf{0.026 \pm 0.011}$ | $0.662 \pm 0.055$ |

Table 5: OOD detection and classification accuracy on the imbalanced CIFAR100 dataset vs. SVHN.

| Architecture | Algorithms | Accuracy | AUROC |
|---|---|---|---|
| ResNet-50 | DNN | $50.630 \pm 0.155$ | $0.747 \pm 0.019$ |
| | Energy-Based (Liu et al., 2021) | $50.630 \pm 0.155$ | $0.772 \pm 0.042$ |
| | SNGP(Liu et al., 2020) | $45.752 \pm 3.375$ | $0.783 \pm 0.041$ |
| | DUQ(Van Amersfoort et al., 2020) | - | - |
| | TUrING Processes | $\mathbf{52.388 \pm 0.568}$ | $\mathbf{0.840 \pm 0.019}$ |
| ResNet-101 | DNN | $49.782 \pm 0.161$ | $0.751 \pm 0.006$ |
| | Energy-Based | $49.782 \pm 0.161$ | $0.793 \pm 0.003$ |
| | SNGP(Liu et al., 2020) | $44.073 \pm 2.186$ | $0.748 \pm 0.018$ |
| | DUQ(Van Amersfoort et al., 2020) | - | - |
| | TUrING Processes | $\mathbf{52.648 \pm 1.027}$ | $\mathbf{0.807 \pm 0.028}$ |

to only 90% of the training labels. These results illustrate the importance of using feature preservation methods that do not oppose the training objective and show that intermediate representations are attractive options for maintaining information of the input distribution.

## D METHOD SUMMARY

We summarize our method in Algorithms 1, and 2. The training phase consists of two steps: we first optimize the model and its internal classifiers jointly using the SDN loss in Equation 5. Second, we derive the individual prediction switches $\mathbf{s}$ and uncertainty scores $\mathbf{v}$, and fit the combination head weights with logistic regression. During prediction, we calculate the final score as a weighted average using the uncertainty weights as well as the internal classifier uncertainty scores $u_s(f^i_{w_i}(\mathbf{x}))$. In our implementation, we calculate the individual uncertainty scores through distance awareness, similar to SNGP (Liu et al., 2020). Applied top our algorithm, the final layer of both the internal classifier, as well as the prediction output are Laplace-approximated Gaussian processes, and we calculate uncertainty with the Dempster-Shafer metric:

$$u_s(\mathbf{x}) = \frac{K}{K + \sum_{k=1}^{K} exp(g^k(\mathbf{x}))} \tag{19}$$

Here, $g^k$ is the k-th logit of the output $g$ (either model prediction or internal classifier), and $K$ represents the number of classes. Our choice is based on simplicity and the past success of Gaussian process layers in single-pass uncertainty estimation (van Amersfoort et al., 2021; Liu et al., 2020). While our design produces sufficient results, we emphasize that other implementations of both $u_s$ and $h_w$ may further improve the performance.

---

**Algorithm 1** Training

1: **Input**
   Labeled Training Set
   $\{\mathbf{x}_i \in \mathcal{X}_{ID}, y_i\}_{i=1}^N$
   Unlabeled Validation Set
   $\{\mathbf{x}_i \in \mathcal{X}_{ID}\}_{i=1}^{N_{val}}$

2: **SDN Training**
3: **for** $epoch \in [1, epochs]$ **do**
4: $\quad h_w \leftarrow$ SGD Update $L_{SDN}$
5: **end for**

6: **Fit Combination Head**
   ▷ Derive prediction switches and scores
   7: $\mathbf{s} \leftarrow \{s(\mathbf{x}_i)\}_{i=1}^{N_{val}}$
   8: $\mathbf{v} \leftarrow \{[u_s(f_{w_1}^1(\mathbf{x}_i)),.., u_s(f_{w_{N_{IC}}}^{N_{IC}}(\mathbf{x}_i))]\}_{i=1}^{N_{val}}$
   ▷ Fit logistic regression weights
   9: $r_1, ..., r_{N_{IC}} = LR(\mathbf{s}, \mathbf{v})$
10: **Return** $h_w, r_1, ..., r_{N_{IC}}$

---

**Algorithm 2** Prediction

1: **Input**
   Test Sample $\mathbf{x}_{te}$

2: **SDN Prediction**
   ▷ Internal uncertainty scores
   3: $\mathbf{u} \leftarrow [u_s(f_{w_1}^1(\mathbf{x}_{te})), ..., u_s(f_{w_{N_{IC}}}^{N_{IC}}(\mathbf{x}_{te}))]$

   ▷ Combine scores
   4: $u_{final} \leftarrow \frac{1}{\sum_{i=1}^{N_{IC}} r_i} \sum_{i=1}^{N_{IC}} r_i u_s(f_{w_i}^i(\mathbf{x}))$

   ▷ Prediction
   5: $\tilde{y}_{te} \leftarrow h_w(\mathbf{x}_{te})$
   6: **Return** $\tilde{y}_{te}, u_{final}$

---

# E    ADDITIONAL RELATED WORK ON DISTANCE PRESERVING NEURAL NETWORKS

The goal of learning a distance-preserving mapping has been an important objective in a wide range of fields such as generative modeling (Lawrence & Quiñonero Candela, 2006; Dinh et al., 2014; 2016) and dimensionality reduction (Abraham et al., 2006; Perrault-Joncas & Meila, 2012). Recently, the concept has been expanded to uncertainty estimation for neural networks and is used to enable single-pass uncertainty estimators (Liu et al., 2020). Several methods exist to control distance preservation in neural networks and each comes with its own set of trade-offs: the two-sided gradient penalty (Gulrajani et al., 2017) was originally introduced in the context of GANs as an alternative to weight clipping (Arjovsky et al., 2017). The penalty regularizes the network by penalizing the squared distance of the gradient from a fixed value for every input point. The approach is popular due to its simple implementation, but represents only a soft constraint. Spectral normalization (Gouk et al., 2021; Miyato et al., 2018) combines spectral normalization with residual connections to implement distance preservation. The method represents a global constraint as it regularizes by normalizing the weights and suffers less from training instabilities as the gradient penalty. Finally, there exists work on reversible networks that force distance preservation through reversible layers and avoiding down-scaling operations (Jacobsen et al., 2018; Behrmann et al., 2019). In practice, reversible models are difficult to train and consume considerably more memory in practice (van Amersfoort et al., 2021). For this purpose, recent single-pass approaches utilize either spectral normalization, or the gradient penalty.