# OpenReview forum: "Transitional Uncertainty with Intermediate Neural Gaussian Processes"
_ICLR.cc/2024/Conference — ICLR 2024 Conference Withdrawn Submission_

### Official Review · Reviewer_3eZi · 2023-10-30

**Soundness:** 3 good
**Presentation:** 2 fair
**Contribution:** 2 fair
**Rating:** 5
**Confidence:** 3

**Summary:**

-	The paper investigates feature-based uncertainty estimation in image classification. The paper analyzes properties of different feature preservation methods and presents motivation for transitional feature preservation, where feature distances are encoded in differences of the individual sample representations. As existing transitional feature preservation methods require multiple passes, the paper proposes TUrING Processes, a single-pass uncertainty estimation method by combining layer-wise hidden representations instead of ensembles. Here, the gaussian process is used to quantify uncertainty. The results show that the proposed method outperforms baselines on OOD detection consistently across different models and datasets although the task performance is sacrificed in some cases.

**Strengths:**

-	The paper investigates an important area of uncertainty estimation.

-	The paper proposes a way to mitigate the limitations of existing single-pass uncertainty estimation methods.

-	The proposed method is effective in OOD detection on image classification.

**Weaknesses:**

-	The exact claim the paper is making is unclear to me as the paper only focuses on OOD detection rather than uncertainty estimation.

-	Some presentational details are not sufficiently clear. For example, Figure 1b compares ensemble workflow and combining layer-wise representations. In the ensemble workflow, uncertainty is measured from differences between several representations (with ensembles) of the same sample. However, the proposed method does not use differences between representations from different layers. Simply combining intermediate layer representations does not work in a similar way to the ensemble workflow.

-	Some claims are not substantiated clearly enough. For example, I am curious whether there is evidence for the claim “the network collapses the class clusters to single points creating a challenging setting for uncertainty estimation,” in the left plot of Figure 1a.

-	The insights are limited despite the effectiveness of the proposed method.

**Questions:**

-	The paper claims that “the network collapses the class clusters to single points creating a challenging setting for uncertainty estimation.” However, in my opinion, it would be challenging for OOD detection, not for uncertainty estimation. Could this point be clarified?

-	It is logical that “feature preservation in the output” may harm the performance. However, why is transitional feature preservation advantageous over “no preservation” for uncertainty estimation?

-	Regarding the issues mentioned above about Figure 1b, it would be good to clarify how the proposed method (or each component) directly solves the discussed problems: (1) feature preservation and (2) efficiency (ensembles).

-	Additional experiments are required to support the main claims. This may include (1) ablation studies to see the effects of each component of the proposed method, and (2) comparisons with existing iterative uncertainty estimation methods in terms of time and space complexity.

---

### Official Review · Reviewer_AwFU · 2023-11-05

**Soundness:** 3 good
**Presentation:** 3 good
**Contribution:** 2 fair
**Rating:** 5
**Confidence:** 3

**Summary:**

The paper studies the problem of _single pass uncertainty estimation_, i.e., being able to estimate epistemic uncertainties of a deep neural network using a single pass during inference. This is different from several existing approaches that either require multiple networks or multiple passes for uncertainty estimation.

The proposed idea called $\mathrm{TUrING~Processes}$  exploits intermediate feature representations of the DNN to estimate uncertainty. The idea behind this is called transitional feature preservation, that ensures there is no feature collapse during training -- which the paper argues is essential for accurate uncertainties.

This is implemented in practice using shallow deep networks (SDNs) through early exits in the network layers, with individual classifiers for each. These predictions are then combined using an additional combination head that is fit in a post-hoc manner using unlabeled validation examples.

There are theoretical and empirical justifications for how the proposed method is effective in feature distance preservation. Experiments on CIFAR10/100 and Medical images show $\mathrm{TUrING~Processes}$ to be effective in rejecting outliers over other baselines.

One side note on the abbreviation  $\mathrm{TUrING~Processes}$ : This name is overloaded (especially in AI), and it can be confusing or misleading. It may be more effective to change it so it reflects the paper's contributions and unique to it.

**Strengths:**

* Single pass uncertainty estimation is an important problem and the paper makes an interesting and novel insight into how intermediate feature representations can be exploited. It is intuitive that (with appropriate weighting) intermediate features do have somewhat complimentary information, and it is interesting to see it being exploited for uncertainties.
	* I am curious how the learned weights are for the combination head ($r_1, r_2...r_N$ ), and if they are some how indicative of (a) task hardness, (b) generalization error prediction. Further, if the validation was done on OOD data, would the distribution of these weights be likely different (since they are somehow measuring the importance of differnet layers on the final task, if i understand correctly)

* The ideas are explained clearly, and easy to follow.
* The experiments show good outlier rejection on CIFAR 10/100 and medical image modalities, most of the time, justifying the use of the proposed method

**Weaknesses:**

* **Architecture**: One of the unacknowledged limitations of single pass UQ methods is the need to modify architectures from the original, which is non-trivial to do. This is especially true in this paper where its required to take early exists from all layers, and fit individual classifiers at each level, followed by a combination head. For smaller datasets like CIFAR10/100 this is relatively easy, but can become tricky when optimizing for larger benchmarks (ImageNet), which are not tested in the paper.

  Further, I dont see the comparison between the baselines to be an apples-apples comparison as this method requires several additional parameters (tens of thousands more?) This is likely only scales as the architectures get larger.. Further, most SoTA are moving towards attention-based models, and for the paper's contribution to be impactful, uncertainty estimation with these architectures must also be studied.
* **Empirical Evaluation**: In addition to evaluations on larger scale benchmarks like ImageNet, more evaluations baselines on outlier rejection approaches are needed -- there are several state of the art -- including ones that do no require outlier validation data.
* **Calibration** : While it is claimed that the proposed method is "preferable" under distribution shifts,  this is never actually evaluated -- how is it preferred? Does it generalize better? How well is it calibrated? I think these also need to be addressed.

**Questions:**

Please see comments above

---

### Official Review · Reviewer_SFMY · 2023-11-07

**Soundness:** 3 good
**Presentation:** 3 good
**Contribution:** 2 fair
**Rating:** 5
**Confidence:** 3

**Summary:**

the paper introduces an approach to uncertainty quantification in deep neural networks through "transitional uncertainty with intermediate neural gaussian processes" (turing processes). the proposed method aims to preserve feature distances in intermediate layers of a dnn. the manuscript comes with theoretical justifications, in sections 3 and 4, and empirical evidence across several datasets, including medical imaging datasets. the authors also provide helpful supplements.

**Strengths:**

1. the paper is well-written, with a clear narrative that guides the reader through the problem statement, proposed solution, and validation.
1. originality of the approach lies in its use of intermediate layer features, as laid out in section 2.2. particularly for applications where single-pass uncertainty estimation is necessary, this provides an interesting approach.
1. the preservation of feature distances is a novel extension to common, existing uncertainty quantification methods.
1. the quality of the experimentation and methodology is thorough.

**Weaknesses:**

1. the biggest gap i see overall is a lack of contextualization wrt to existing work on single pass uncertainty quantification. while i have full compassion that we can never benchmark against everything, even in more niche problem domains as uncertainty quantification, critical works should be included or their omission at least explained. among these i would count gast's probout which also offers a layerwise propagation version (https://openaccess.thecvf.com/content_cvpr_2018/html/Gast_Lightweight_Probabilistic_Deep_CVPR_2018_paper.html), quantile regression (https://www.jstor.org/stable/1913643) and conformal prediction (https://www.jmlr.org/papers/volume9/shafer08a/shafer08a.pdf), interval neural networks (https://arxiv.org/abs/2003.11566) as well as the classic direct variance prediction (https://proceedings.neurips.cc/paper/1994/hash/061412e4a03c02f9902576ec55ebbe77-Abstract.html). regarding naming of the method, a mention of relation to evidential turing processes https://arxiv.org/abs/2106.01216 from last year's iclr also appears opportune.
1. the supplementary material provides a good depth of technical detail, but integrating some of this content into the main body could make the paper more self-contained, especially the algorithm description and runtime considerations.
1. the paper could benefit from a more detailed discussion on computational efficiency, particularly in section 5 where the comparison to ensemble methods lacks a thorough analysis of time complexity.
1. generalizability is touched upon, but the paper could include additional experiments or a theoretical discussion on how turing processes might perform with different architectures or in other domains.
1. hyperparameter sensitivity is not addressed in depth; section 4 could be expanded to include a discussion on the methodology for selecting hyperparameters and their impact on the model's performance.

**Questions:**

1. in section 5, the computational overhead of turing processes is compared to ensemble methods. could the authors provide more detailed benchmarks, including time complexity and resource utilization metrics?
1. the paper presents promising results for the tested architectures and domains. how do the authors envision the adaptation of turing processes to other types of neural networks and problem domains?
1. hyperparameter tuning is critical for model performance. can the authors elaborate on the process used to select hyperparameters for turing processes, as mentioned briefly in section 4.3?
1. what is the relation to other works i mentioned above? could you provide some additional contextualization?

---

> ### Comment · Reviewer_SFMY · 2023-11-21
> **discussion period**
>
> no replies or responses were received by the authors.
>
> i thus maintain my rating.
>
> in good spirits,
>
> reviewer SFMY

---

### Author Response · Authors · 2023-11-23
**Withdrawal**

We thank the reviewers for their time and helpful feedback. We note that the feedback includes several experiments that require a lot of time to complete. For this purpose, we will add empirical results and re-submit our work when the experiments are finished.

Kind regards,
the authors